Corrected: Publisher correction

# Dynamic structural states of ClpB involved in its disaggregation function

Takayuki Uchihashi [1], Yo-hei Watanabe[2,3], Yosuke Nakazaki[2,3], Takashi Yamasaki[2,3], Hiroki Watanabe[4], Takahiro Maruno[5], Kentaro Ishii[6], Susumu Uchiyama [5,6], Chihong Song[7], Kazuyoshi Murata[7], Ryota Iino [8,9] & Toshio Ando [10]

The ATP-dependent bacterial protein disaggregation machine, ClpB belonging to the AAA+ superfamily, refolds toxic protein aggregates into the native state in cooperation with the cognate Hsp70 partner. The ring-shaped hexamers of ClpB unfold and thread its protein substrate through the central pore. However, their function-related structural dynamics has remained elusive. Here we directly visualize ClpB using high-speed atomic force microscopy (HS-AFM) to gain a mechanistic insight into its disaggregation function. The HS-AFM movies demonstrate massive conformational changes of the hexameric ring during ATP hydrolysis, from a round ring to a spiral and even to a pair of twisted half-spirals. HS-AFM observations of Walker-motif mutants unveil crucial roles of ATP binding and hydrolysis in the oligomer formation and structural dynamics. Furthermore, repressed and hyperactive mutations result in significantly different oligomeric forms. These results provide a comprehensive view for the ATP-driven oligomeric-state transitions that enable ClpB to disentangle protein aggregates.

[1] Department of Physics and Structural Biology Research Center, Nagoya University, Chikusa-ku, Nagoya 464-8602, Japan. [2] Department of Biology, Faculty of Science and Engineering, Konan University, Okamoto 8-9-1, Kobe 658-8501, Japan. [3] Institute for Integrative Neurobiology, Konan University, Okamoto 8-9-1, Kobe 658-8501, Japan. [4] Department of Physics, College of Science and Engineering, Kanazawa University, Kanazawa 920-1192, Japan. [5] Department of Biotechnology, Graduate School of Engineering, Osaka University, Osaka 565-0871, Japan. [6] Exploratory Research Center on Life and Living Systems (ExCELLS), National Institutes of Natural Sciences, Okazaki, Aichi 444-8787, Japan. [7] National Institute for Physiological Sciences, National Institutes of Natural Sciences, Okazaki, Aichi 444-8787, Japan. [8] Institute for Molecular Science, National Institutes of Natural Sciences, Okazaki, Aichi 444-8787, Japan. [9] Department of Functional Molecular Science, School of Physical Sciences, The Graduate University for Advanced Studies (SOKENDAI), Hayama, Kanagawa 240-0193, Japan. [10] Nano Life Science Institute (WPI-NanoLSI), Kanazawa University, Kanazawa 920-1192, Japan. These authors contributed equally: Takayuki Uchihashi, Yo-hei Watanabe. Correspondence and requests for materials should be addressed to Y.-h.W. (email: ywatanab@center.konan-u.ac.jp) or to R.I. (email: iino@ims.ac.jp) or to T.A. (email: tando@staff.kanazawa-u.ac.jp)

The superfamily of ATPases associated with diverse cellular activities (AAA+) includes a wide variety of members that are involved in membrane fusion, DNA replication, protein degradation, and others[1,2]. Most of the members form ring shaped hexamers, bind macromolecule substrates, and remodel the substrates by using their ATP-fueled conformational changes. Despite the diversity of functions, AAA+ proteins are thought to share a common mechanism, attracting much attention. However, our knowledge about their structural dynamics, which is the key to understanding the mechanism of action of these proteins, has been limited[3].

Bacterial ClpB, a member of the AAA+ family, and its yeast homologue Hsp104 are protein disaggregases, contributing to cell survival under heat stress conditions[4–9]. ClpB consists of a globular N-terminal domain, two AAA+ modules (AAA1 and AAA2), and a long coiled-coil M-domain (MD)[10]. The N-terminal domain is highly mobile and participates in substrate binding but is not essential for the disaggregation activity[11–13]. Both AAA1 and AAA2 bind and hydrolyze ATP and constitute a core of the hexameric ring[10,14,15]. It has been assumed that, during the disaggregation reaction, aggregated proteins are threaded through the central pore of the ring[16]. Recently, the oligomeric structures of ClpB and Hsp104 in which the central pores were occupied by a model denatured protein (casein) were shown by cryo-electron microscopic (cryo-EM) single particle analysis[17,18].

The AAA1 and AAA2 modules contain the Walker A and B consensus motifs that are critical for ATP binding and hydrolysis, respectively. Individual roles of these AAA+ modules in ClpB have been studied by introducing mutations to the Walker motifs. ATP-binding to AAA1 but not to AAA2 is required for the formation of a stable oligomer[14,15,19]. A mutant defective in ATP hydrolysis but not ATP binding in both AAA+modules stably binds substrate proteins[20]. The AAA1 and AAA2 modules regulate each other in a complex manner as regards ATP-binding and hydrolysis[21–25], and the cooperative ATPase cycle causes structural changes to drive the threading of aggregated proteins[12,17,18], although the threading of denatured proteins does not require ATP hydrolysis[26]. The MD protruding from the middle of AAA1 surrounds the ClpB ring and regulates both ATPase and disaggregation activities of ClpB[10,12,27–30].

For full disaggregation activity, ClpB requires DnaK co-chaperone containing two functional domains, the nucleotide binding domain (NBD) and the substrate binding domain (SBD)[31,32]. DnaK bound to aggregated proteins interacts with the MD of ClpB at the NBD, and this interaction unleashes ClpB from its repressed state and activated ClpB threads the aggregated proteins recruited by DnaK[33,34]. However, independently of DnaK, the MD-originated repression can be canceled or maintained by MD mutations, resulting in hyperactive or repressed ClpB, respectively[27,30,35,36]. According to the single particle cryo-EM study of ClpB, the MDs in the repressed mutant interact with each other in a head-to-tail manner, forming a coil around the AAA1 ring, whereas in the hyperactive mutant, the interaction between MDs is disrupted and thereby the ring tightening is relieved[28].

Although most proposed models for disaggregation by ClpB/Hsp104 are based on their hexameric ring structure, the structure is not so robust. The ringed oligomer formation depends on many factors, such as nucleotide binding, temperature, and protein and salt concentrations[14,15,37]. The ring easily collapses and their subunits are exchanged, even in the middle of the disaggregation reaction[22,38]. It was proposed that the ring fragility would contribute to avoiding jam of aggregated protein during threading[38]. In addition, a recent cryo-EM study demonstrated that Hsp104 forms a two-turn spiral in which AAA1 in one

protomer is linked to AAA2 in an adjacent protomer[17,39]. Thus, understanding the dynamic nature of oligomeric structure is essential for elucidating the disaggregation mechanism of ClpB.

In this study, we directly observed the dynamic structure of ClpB using high-speed atomic force microscopy (HS-AFM) that can visualize proteins in action at high spatiotemporal resolution[40–42]. HS-AFM images of ClpB reveal highly diverse and transient oligomeric forms exhibiting round and distorted closed rings and even open rings. Analysis of dynamic transitions between these oligomeric forms demonstrates that the round hexameric ring is stabilized by ATP binding, whereas ATP hydrolysis provokes massive changes in the ring structure. HS-AFM analysis of Walker motif mutants of ClpB provides an insight into the respective roles of ATP binding and hydrolysis occurring in AAA1 and AAA2 in the formation and dynamic changes of oligomeric structures. Notable differences in the oligomeric forms between the repressed and hyperactive states suggest a critical role of destabilized inter-subunit interactions in the chaperone activity of ClpB.

## Results

**Oligomeric forms of TClpB depending on ATP concentration.** Since ATP binding to ClpB is essential for its oligomer formation[15], *T. thermophilus* ClpB (*T*ClpB) was preincubated in the presence of 2 mM ATP at 55 °C for 1 min. Then, the sample was adsorbed onto bare mica surface for HS-AFM imaging in 0.5 mM ATP at 25 °C. Full-length *T*ClpB appeared as incomplete rings or circular particles with a high peak at their center (Supplementary Figure 1a). The latter species are possibly closed rings but their central pore is unresolvable due to the N-terminal domain locating in the close proximity to the pore. In fact, HS-AFM images of the N-terminal deletion mutant of *T*ClpB (ΔN-*T*ClpB) preincubated in the same way as above showed rings with a central pore (Fig. 1a). This appearance difference between the full-length and ΔN-*T*ClpB strongly suggests that the C-terminal AAA2 preferentially faces mica under the buffer condition used in this study. The orientation was further confirmed by binding of streptavidin to the top surface of ΔN-*T*ClpB biotinylated at the N-terminal side (ΔN-*T*ClpB-Q142C-biotin) (Supplementary Figure 1b−d). Since the N-terminal deletion does not have a major impact on the disaggregation activity of *T*ClpB[11–13] (Supplementary Figure 2), we hereafter use the ΔN-*T*ClpB as wild-type.

The oligomers formed by ΔN-*T*ClpB showed highly diverse structures in the presence of 0.5 mM ATP, which could be classified into four typical forms (Fig. 1b–e). One form is a round hexameric ring, although the resolution of HS-AFM lower than X-ray crystallography and cryo-EM single-particle analysis prevented us from judging if the ring is really symmetrically round (Fig. 1b and Supplementary Figure 3a). In one of distorted ring forms, the ring contains a prominent protrusion and a seam next to the protrusion (Fig. 1c and Supplementary Figure 3b), similar to those observed in cryo-EM images of Hsp104 assembled into an asymmetric spiral hexamer (Fig. 1f)[39]. When a simulated AFM image (Fig. 1g) was constructed using the spiral structure of Hsp104 (PDB code: 5KNE) from which the N-terminal domains were truncated, it resembled the observed HS-AFM image of ΔN-*T*ClpB (Fig. 1c). We thus concluded that the AFM image shown in Fig. 1c corresponds to the left-handed spiral hexamer similar to that observed for Hsp104. Furthermore, another distorted ring was also observed, which has two protrusions at the opposite sites in the ring (Fig.1d) with six height peaks (Supplementary Figure 3c). This ring appeared to consist of a pair of half spirals coalesced with each other in a head-to-tail arrangement. This view was reinforced by observed

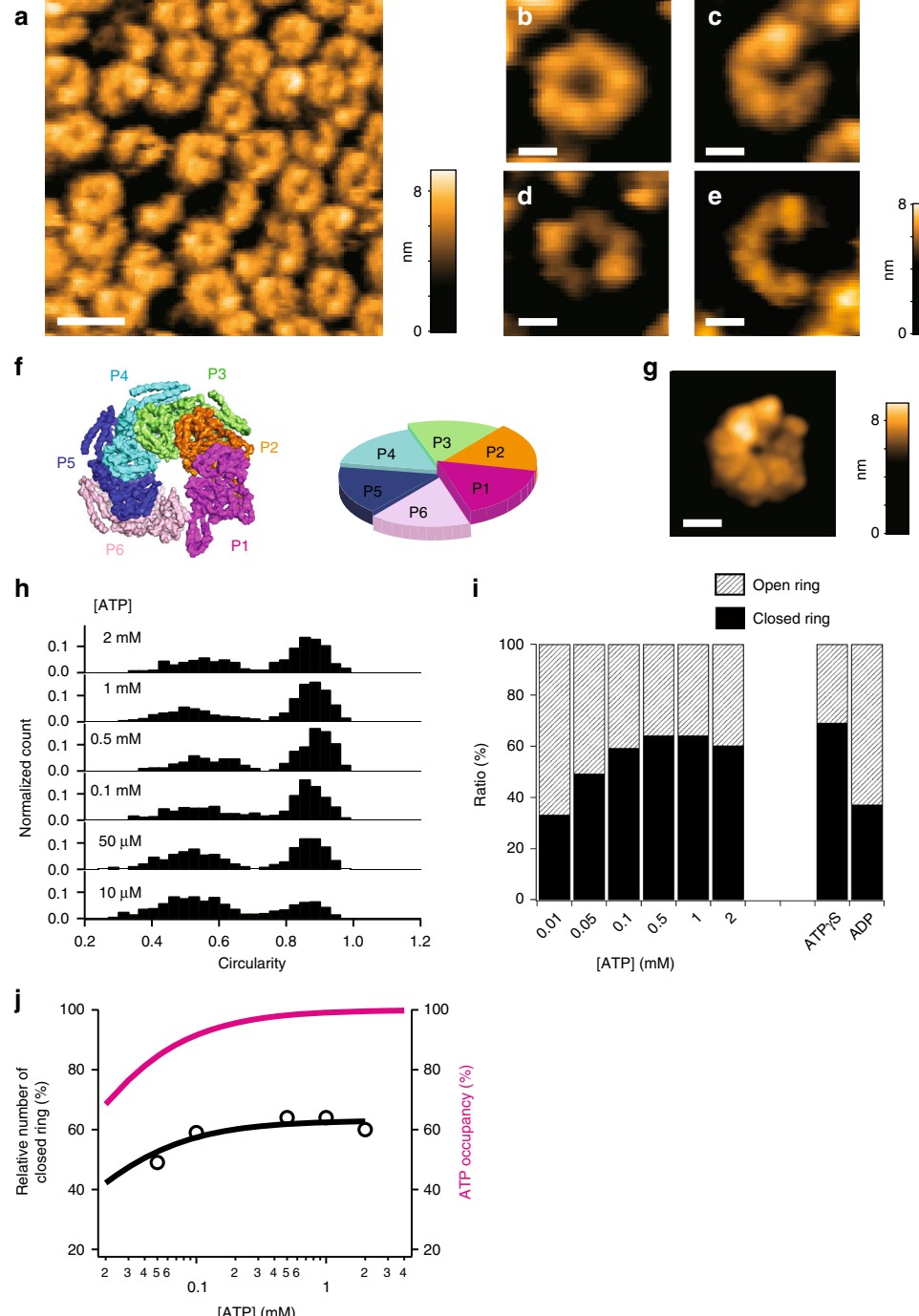

**Fig. 1** Diverse oligomeric forms of ΔN-*T*ClpB observed by HS-AFM. **a** HS-AFM image acquired in 0.5 mM ATP at 25 °C. Scale bar, 20 nm. Z color bar, 0 to 9.2 nm. **b**–**e** Four typical oligomeric forms observed in the presence of ATP: **b** round closed ring, **c** spiral, **d** twisted-half-spiral, and **e** open ring. We collectively refer to the spiral and twisted-half-spiral rings as the distorted closed ring. Scale bar, 5 nm. Z color bar, 0 to 8.0 nm. **f** Cryo-EM structure of spiral hexamer of Hsp104 without N-terminal domains (PDB code: 5KNE)[39] and schematic of the spiral architecture. **g** Simulated AFM image of N-terminal-truncated spiral hexamer of Hsp104 constructed using a cone-shaped probe with a tip radius of 0.5 nm and a half cone angle of 5°. The constructed image was filtered by a low-pass filter with a cut-off frequency of 3 nm. Scale bar, 5 nm. Z color bar, 0 to 9.3 nm. **h** Histograms of circularity for the oligomers formed at different [ATP]. Numbers of molecules analyzed are 425 (10 μM), 389 (50 μM), 438 (0.1 mM), 415 (0.5 mM), 624 (1 mM), and 410 (2 mM). **i** Percentages of closed (black area) and open (hatched area) forms at various [ATP] estimated from the histograms. A threshold value of circularity was set at 0.68. Percentages of the closed and open forms observed at 1 mM ATPγS and 1 mM ADP are also displayed (413 molecules for ATPγS and 432 molecules for ADP). **j** Percentage of closed rings as a function of [ATP] (open circles). The data points were fitted by a simple 1:1 binding model with an apparent $K_d = 9.9 \pm 1.8$ μM (mean ± s.d.) (black line). The magenta line shows ATP occupancy in the ATP-binding sites of ΔN-*T*ClpB hexamers theoretically calculated using an apparent $K_d$ (9.2 μM) estimated by replacement titration of bound mant-ADP with ATP (Supplementary Figure 8)

dynamic changes of this ring, as described later. We hereafter refer to this form as the twisted-half-spiral form. One possible structural model of this form is that the interactions between the protomers 1 and 6, and 3 and 4 are disrupted, resulting in a ring formed by the two trimer units (P1-3 and P4–6), in which the height of these trimer units descends from P1 to P3 and from P4 to P6 (Supplementary Figure 3e). The overall feature of this form is very similar to that of asymmetric hexamers found in another AAA+ protein-unfolding machine[43], ClpX, potentially implying a common unfolding mechanism. The forth oligomeric form has a worm-like shape (Fig. 1e), which is likely to be an intermediate towards the round closed ring or the spiral form. As detected with HS-AFM, both open and closed rings were observed with negative-staining electron microscopy for both full-length and ΔN-TClpB (Supplementary Figures 4 and 5). These results indicate that the open form is an inherent structure of the full-length ClpB and not an artifact caused by deletion of the N-terminal domain. The number of protomers containing in the worm-like form shown in Fig. 1e was six judging from the number of height peaks (Supplementary Figure 3d), although the number of protomers varied ranging from three to eight (Supplementary Figure 6).

The first three forms including the round closed ring, the spiral and twisted-half-spiral basically represent the closed rings, while the worm-like form is an open ring in which an inter-protomer connection is completely disrupted. These forms were observed to interchange dynamically in the presence of ATP, as described later. The two representative states (closed- and open-ring states) can be quantitatively distinguished by measuring circularity (Supplementary Figure 7a). The histograms of circularity at different ATP concentrations clearly showed two peaks (Fig. 1h). The lower and higher circularity peaks correspond to the open- and closed-ring states, respectively. The relative frequencies of occurrence of the two ring states were estimated by dividing the whole histogram area into two areas at a threshold circularity value of 0.68 (Fig. 1i). The percentage of appearance of closed rings increased with increasing ATP concentration ([ATP]) and saturated around 60% at high [ATP]. The [ATP] dependence of the percentage could be fitted by a simple 1:1 binding model with an apparent $K_d = 9.9 \pm 1.8$ μM (Fig. 1j, open circles), which was in good agreement with the biochemically estimated apparent $K_d$ value (9.2 μM) for ATP binding to ΔN-TClpB (Fig. 1j, magenta line, and Supplementary Figure 8).

In the presence of 1 mM ATPγS, which is a poorly hydrolysable ATP analogue for TClpB, the percentage of the closed-ring state reached almost 70%, whereas it was significantly reduced to about 30% in the presence of 1 mM ADP (Fig. 1i). The overall structure of the closed ring in the presence of ATPγS was similar to that observed in cryo-EM images of ClpB reported recently[18]. The center value of the circularity distribution for the closed ring under the ADP condition was slightly lower than that measured under the ATPγS condition (Supplementary Figure 7b). Furthermore, the distribution under the ADP condition was broader than that of ATPγS condition, indicating that most of the closed rings are in distorted forms under the ADP condition. These results suggest that ATP binding to TClpB induces the round closed-ring form, whereas in the ADP-bound state the ring is distorted.

To quantitatively analyze the number of protomers in an oligomer of full-length or ΔN-TClpB in solution, we carried out sedimentation velocity analytical ultracentrifugation (SV-AUC)[44] (Supplementary Figure 9). Interestingly, we found that trimers are major and hexamers and other oligomers are minor for both full-length and ΔN-TClpB in the presence of 20 mM ATP (Supplementary Figure 9a–c, red lines). This result is consistent with the twisted-half-spiral form consisting of two trimer units.

On the other hand, in the presence of 2 mM ATPγS, the fraction of hexamers is significantly increased for both full-length and ΔN-TClpB (Supplementary Figure 9a and b, blue lines). In SV-AUC, complete sedimentation required for data analysis takes long time (several hours), and ATP is gradually hydrolyzed into ADP by TClpB. This gradual decrease of ATP (gradual increase of ADP) may result in a low fraction of larger oligomers. To confirm this notion, we also carried out measurements in the presence of 20 mM ADP (Supplementary Figure 9a and b, green lines). By the measurements, we found that monomers and oligomers smaller than hexamers were again dominant for both full-length and ΔN-TClpB, although their distributions were different from those in 20 mM ATP and trimers were not major. In all nucleotide conditions tested, we also found a tendency that ΔN-TClpB shows larger oligomers than full-length TClpB, suggesting that truncation of the N-terminal domain slightly enhances the oligomer formation.

In addition to SV-AUC, we also carried out a native mass spectrometry (nMS) analysis[45] (Supplementary Table 1 and Supplementary Figure 10). As results, monomers and oligomers from dimers to hexamers were detected for both full-length and ΔN-TClpB in the presence of 0.1 mM ATP or ATPγS. The largest oligomer detected by nMS was hexamers, suggesting that oligomers larger than hexamers do not form stable rings. Furthermore, in contrast to the results of SV-AUC, the mass spectra were almost identical in the presence of ATP and ATPγS. This is likely to result from ring disassembly during the process of ionization in nMS.

**Conformational dynamics of ΔN-TClpB oligomers.** Figure 2a shows clipped images of ΔN-TClpB recorded at 10 fps in the presence of 10 μM ATP. The twisted-half-spiral form that appeared in the first frame was converted to the spiral form at 4.4 s, then returned back to the twisted-half-spiral form at 6.8 s. One of the two seams in the twisted-half-spiral, and the single seam in the spiral transiently broke at 8.1 s and 10.0 s, respectively. The round closed ring appeared at 11.3 s, which was followed by the spiral and twisted-half spiral forms. As seen here, the oligomeric structure of ΔN-TClpB is not static but undergoes dynamic transitions among different forms (Supplementary Movies 1–5). Structural fluctuations and seam breaks in the twisted-half spiral forms observed in these movies indicate that each half spiral behaves as a unit, although three protomers composing one half spiral are occasionally altered after a round of disappearance and reappearance of a twisted-half-spiral. The change of two protruded positions also indicates that each protomer can carry out up-and-down motion.

To assess whether or not the structural transitions observed above are indeed driven by ATP hydrolysis, we analyzed the time course of occurrence of different oligomeric forms and its [ATP] dependence. To perform semi-automatic discrimination between the round and distorted closed rings, we used the standard deviation of cross-sectional height variations (SDCH) measured along the top surface of the ring (Supplementary Figure 11). The SDCH value of 0.3 nm was used as a threshold of the two states of ring structures; SDCH less than 0.3 nm corresponds to the round state, while SDCH larger than 0.3 nm corresponds to the distorted one. The frequency of transitions between the round and distorted closed rings was low at low [ATP] (Fig. 2b, bottom panel), and increased with increasing [ATP] (Fig. 2b, middle and upper panels, and Supplementary Figure 12).

The histograms of SDCH for different [ATP] clearly showed that the number of image frames (i.e., total appearance time) that captured the round or distorted rings is [ATP]-dependent (Fig. 2c, lower five panels). At 10 μM ATP, the SDCH distribution was

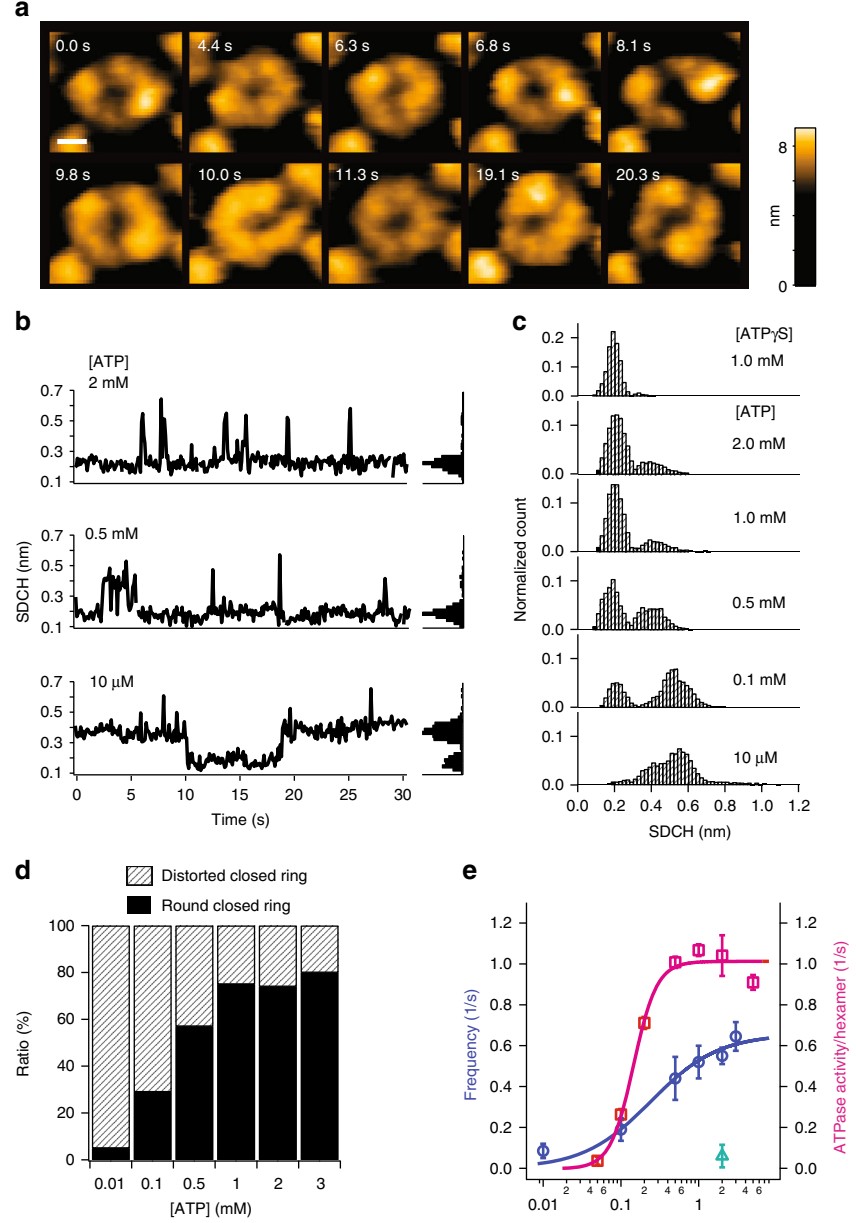

**Fig. 2** Conformational dynamics of ΔN-*T*ClpB. **a** Clipped HS-AFM images from those successively captured at 10 fps in the presence of 10 μM ATP. Scale bar, 5 nm. Z color bar, 0 to 9.1 nm. **b** Typical time courses of SDCH measured along the hexamers in the presence of different [ATP] (left) and corresponding histograms of SDCH (right). **c** Histograms of SDCH measured at various [ATP] and 1 mM ATPγS. **d** Percentages of round (black area) and distorted (hatched area) closed forms at various [ATP]. **e** ATPase activity (magenta open squares) and frequency of structural transition cycles between the round and distorted closed forms (blue open circles) as a function of [ATP]. The ATPase activity was fitted by Hill's equation with the Hill coefficient of $2.6 \pm 0.6$ and the $K_m$ of $0.16 \pm 0.02$ mM (means ± s.d., magenta line). The transition frequency was well fitted by the Michaelis–Menten equation with $K_m = 0.35 \pm 0.13$ mM (mean ± s.d., blue line). A green open triangle indicates the frequency observed at 2 mM ATPγS

broad and its main peak appeared around 0.58 nm, indicating that the oligomers were in the distorted state during most of the observation time. With increasing [ATP], the SDCH distribution component with a peak around 0.2 nm became higher (Fig. 2c), indicating that the total appearance time of round closed rings becomes longer at higher [ATP] (Fig. 2d); even when the oligomers often undergo round-to-distorted state transitions, they are quickly backed to the round state at higher [ATP].

To understand the relationship between the [ATP] dependence of the transition frequency and the ATPase activity, ATPase activities of ΔN-*T*ClpB at various [ATP] were spectrometrically

measured using an ATP-regenerating system (Fig. 2e). Consistent with a previous study[21], the ATPase activity of ΔN-*T*ClpB showed positive cooperativity. The curve of ATPase activity against [ATP] was best fitted with the Hill coefficient of $2.6 \pm 0.6$ and $K_m$ of $0.16 \pm 0.02$ mM (Fig. 2e, magenta open squares). However, the frequency of transitions between the round and distorted states measured at various [ATP] was rather best fitted with a simple Michaelis–Menten equation with $K_m = 0.35 \pm 0.13$ mM (Fig. 2e; blue open circles); here the transition frequency is defined by the total number of transition cycles between round and distorted states divided by the total period of HS-AFM

observation. This inconsistency between these [ATP] dependencies would be possibly due to with and without preincubation of the sample in a high [ATP] solution before actual measurements. In the ATPase activity measurements particularly at low [ATP], the sample should not have oligomers. Therefore, the observed cooperativity in the ATPase activity is likely to be caused by different amounts of oligomers formed during the measurements. Consistent with this notion, when ATP was added during HS-AFM imaging of ΔN-$T$ClpB without ATP preincubation, we observed gradual formation of oligomer rings (Supplementary Figure 13 and Movie 6). Hence, the state transitions observed in HS-AFM imaging are possibly well correlated with an ATPase activity that would be innate in mature oligomers of $T$ClpB.

Taken together, $T$ClpB undergoes large-scale dynamic changes in the oligomeric state, driven by the cycles of ATP binding and hydrolysis. Even events of ring disassembly into two separated parts and subunit reshuffling between oligomers were also observed (Supplementary Movies 7 and 8), as suggested from a biochemical study[22]. The dynamic nature of $T$ClpB oligomers was also confirmed by our SV-AUC analysis (Supplementary Figure 9). These dynamic changes would underlie the chaperone activity of ClpB/Hsp104 to disentangle protein aggregates.

**Characterization of Walker motif mutants**. Both AAA1 and AAA2 modules in ClpB contain the Walker A and B motifs. To gain an insight into the involvement of each AAA module in the oligomer formation and dynamic structural changes of $T$ClpB, we performed HS-AFM imaging of its Walker motif mutants. We prepared the following mutants of ΔN-$T$ClpB: Walker A motif double mutants of AAA1 (K204A/T205A abbreviated to 1KT/AA) and AAA2 (K601A/T602A abbreviated to 2KT/AA), Walker B motif mutants of AAA1 (E271Q abbreviated to 1E/Q) and AAA2 (E668Q abbreviated to 2E/Q), and mutants in both AAA1 and AAA2 (i.e., 1KT/AA-2KT/AA and 1E/Q-2E/Q). We first characterized the ATPase and disaggregase activities of these mutants. In the 1KT/AA and 2KT/AA mutants, both ATPase and disaggregation activities were severely inhibited (Supplementary Figures 14a and b). On the other hand, the 1E/Q mutant exhibited an increased ATPase activity, while the 2E/Q mutant showed a slightly decreased ATPase activity (Supplementary Figure 14a). Regarding the disaggregation, the 1E/Q and 2E/Q mutants exhibited activities reduced by 80–90% (Supplementary Figure 14b). These features are consistent with previous studies[15,25].

All the mutants were preincubated in the presence of 2 mM ATP at 55 °C for 1 min, and then HS-AFM observations were carried out in the presence of 1 mM ATP at 25 °C, unless otherwise stated. The 1KT/AA mutant showed inhomogeneous and atypical oligomeric forms (Fig. 3a), consistent with the previous reports that nucleotide binding to AAA1 is essential for the assembly of ClpB[14,15,19]. On the other hand, the 2KT/AA mutant showed ring structures (Fig. 3a and b, and Supplementary Figure 14c), indicating that nucleotide binding to AAA2 is not essential for the ring formation, although no nucleotide binding to AAA2 results in a lower frequency (~50%) of closed ring formation (Fig. 3b). The double Walker A mutant 1KT/AA-2KT/AA showed an appearance very similar to that of 1KT/AA (Supplementary Figure 14d). As for the Walker B mutants, both 1E/Q and 2E/Q mutants formed closed rings at ~70% (Fig. 3a, b, Supplementary Figure 14c). However, their structural features were significantly different. The closed rings formed by the 1E/Q mutant appeared similar to those formed by the ΔN-$T$ClpB and contained both round and distorted forms. On the other hand, the closed rings of 2E/Q mutant homogenously appeared as

round rings. The double Walker B mutant 1E/Q-2E/Q displayed exactly the same appearance as the 2E/Q mutant (Supplementary Figure 14d).

Interestingly, although the 2KT/AA mutant formed closed rings, their size appeared to be widely distributed and some rings were obviously larger compared to those of ΔN-$T$ClpB. For quantitative comparison of the size of closed rings, we measured the contour length along each ring formed by ΔN-$T$ClpB and 2KT/AA (Fig. 3c). The center value and width of contour length distribution for ΔN-$T$ClpB were 35.2 nm and 4.9 nm, respectively. As compared with ΔN-$T$ClpB, the 2KT/AA mutant showed a significantly broader distribution with a center value of 42.1 nm and a width of 9.0 nm. This is due to a large variation in the number of subunits forming ring-shaped oligomers and a large ring distortion caused by loose packing of neighboring subunits (Supplementary Figure 15). Thus, nucleotide binding to AAA2 regulates the number of subunits to be included in $T$ClpB oligomers and is essential for the formation of hexameric rings, although it is not critical for ring formation.

Next, we monitored conformational dynamics of the 2KT/AA, 1E/Q and 2E/Q mutants (Fig. 3d, e). The conformational dynamics of 1E/Q was similar to those of ΔN-$T$ClpB. In fact, the SDCH distribution of the 1E/Q mutant showed two peaks corresponding to the round and distorted states (Fig. 3e). The transition frequency was $0.28 \pm 0.06\,\mathrm{s}^{-1}$, which was almost half the value for ΔN-$T$ClpB at the same [ATP] (1 mM), although the ATPase activity of the 1E/Q mutant is much higher than that of ΔN-$T$ClpB (Supplementary Figure 14a). The rings formed by the 2E/Q mutant were round and static, and did not show conformational transitions during observation (Fig. 3d, e). The 2KT/AA also showed two peaks in the SDCH distribution. However, the distribution was broader than those of ΔN-$T$ClpB and 1E/Q mutant, and no clear border between two states was observed. This may indicate that the 2KT/AA rings undergo conformational changes but the transitions occur gradually without bimodal height changes (Fig. 3e).

From results obtained with all Walker motif mutants, we conclude that the assembly of $T$ClpB monomers into the ring oligomers is driven by nucleotide binding to AAA1, while nucleotide binding to AAA2 governs the correct assembly into hexameric rings. ATP hydrolysis in AAA2 drives large-scale conformational changes of the $T$ClpB ring. The inconsistency between the structural transition rate and the ATPase activity of the 1E/Q mutant suggests that futile ATP hydrolysis partially takes place in this mutant, without driving conformational changes.

**Conformational states of activity mutants**. *E. coli* ClpB (*E*ClpB) harboring single mutations in one of the two ends of the MD coiled-coil, E432A and Y503D, are repressed and hyperactive mutants, respectively[27,28]. The repressed mutant *E*ClpB-E432A lacks the DnaK-dependent disaggregation activity, whereas the hyperactive *E*ClpB-Y503D has higher ATPase and substrate unfolding activities even without Hsp70 partners but abolishes the disaggregation ability due to the loss of interaction with Hsp70 partners[27]. We prepared corresponding repressed (E423A) and hyperactive (Y494D) mutants of ΔN-$T$ClpB (Fig. 4a), which exhibited modified activities in both ATPase and disaggregation activities (Fig. 4b) as similar to the corresponding *E*ClpB mutants. Although the Y494D mutant lost the DnaK-dependent disaggregation activity, the intrinsic disaggregation activity of this mutant was recently demonstrated to be high by using covalently fused yellow fluorescent protein aggregates as substrate[34].

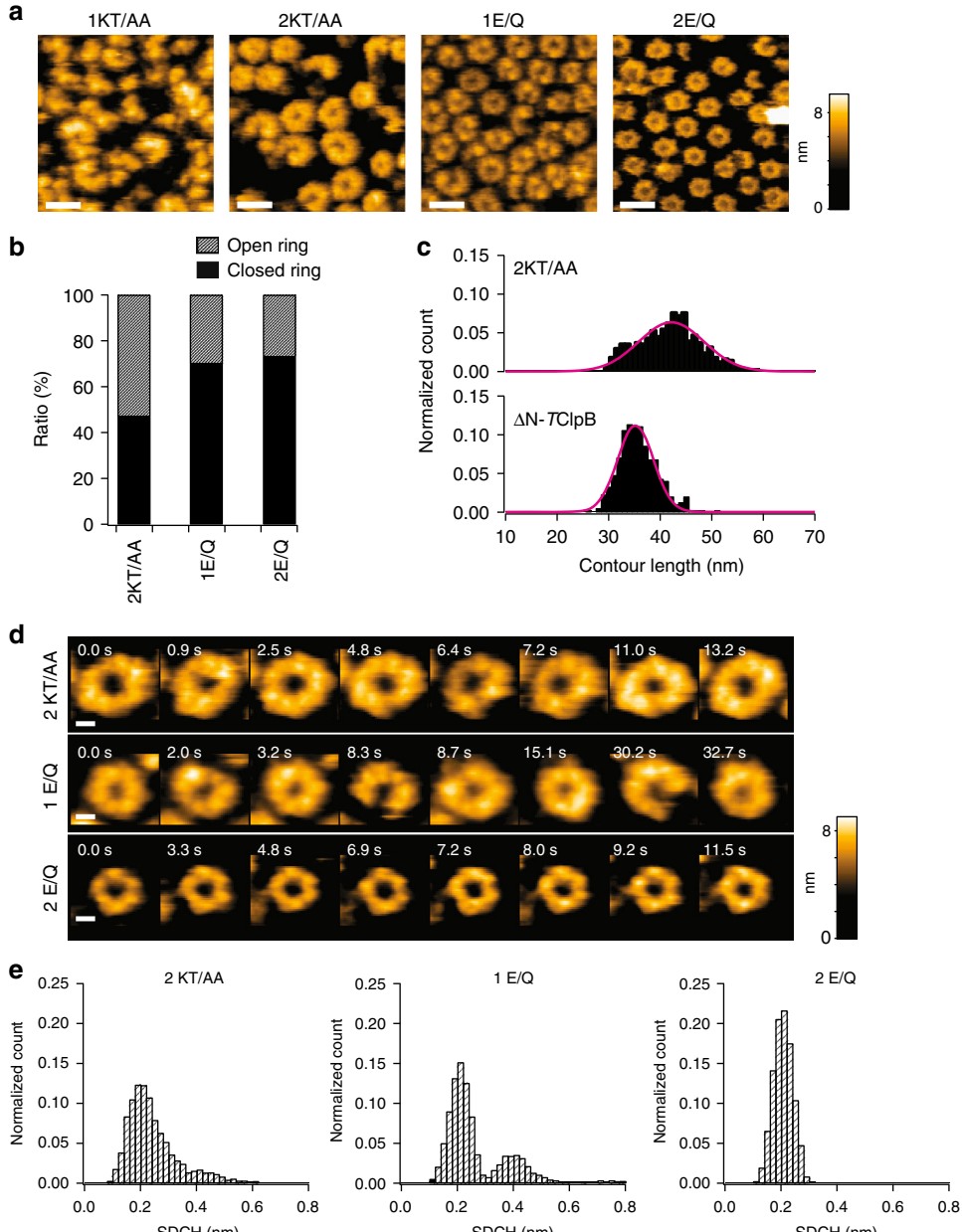

**Fig. 3** Oligomeric forms of Walker motif mutants. **a** HS-AFM images of Walker A (1KA/AA and 2KT/AA) and Walker B (1E/Q and 2E/Q) mutants observed in the presence of 1 mM ATP. Scale bars, 20 nm. Z color bar, 0 to 9.6 nm. **b** Percentages of closed (black area) and open (hatched area) forms of 2KT/AA (1102 molecules), 1E/Q (388 molecules) and 2E/Q (516 molecules). **c** Histograms of contour length along the closed rings of ΔN-TClpB (397 molecules) and 2KT/AA (520 molecules). The center values and widths fitted by Gaussians are 35.2 nm and 5.3 nm (ΔN-TClpB), respectively, and 42.1 nm and 9.0 nm (2KT/AA), respectively. **d** Clipped HS-AFM images of 2KT/AA, 1E/Q, and 2E/Q mutants in the presence of 1 mM ATP. Scale bars, 5 nm. Z color bar, 0 to 9.1 nm. **e** Histograms of SDCH along the rings of 2KT/AA (3543 images from 32 molecules), 1E/Q (5279 images from 9 molecules) and 2E/Q (7287 images from 9 molecules). The frequencies of transition cycle between round and distorted closed states of 1E/Q and 2E/Q mutants are 0.28 ± 0.06 s$^{-1}$ and 0.022 ± 0.002 s$^{-1}$ (means ± s.d.), respectively

These repressed E423A and hyperactive Y494D mutants were preincubated in 2 mM ATP at 55 °C for 1 min, and observed with HS-AFM in 1 mM ATP at 25 °C. The repressed E423A mutant showed both closed and open rings (Fig. 4c and e, and Supplementary Figure 16a), as with the case of ΔN-TClpB at high [ATP] (Fig. 1h and i). However, the SDCH distribution of the oligomers indicates that these closed rings are mostly in the round state (Fig. 4f). The frequency of transitions between round and distorted closed rings was 0.10 ± 0.10 s$^{-1}$ (Supplementary Figure 16b), which was nearly the basal level of ΔN-TClpB

measured in the presence of ATPγS (Fig. 2e), although the ATPase activity of E423A was reduced only by half of that of ΔN-TClpB (Fig. 4b). This suggests that ATP is futilely hydrolyzed without being used to drive dynamic structural changes, similar to the case of the 1E/Q mutant.

Strikingly, the hyperactive Y494D mutant exhibited mostly open-ring oligomers (Fig. 4d) and a small amount of closed rings (Fig. 4e). Most of open rings were hexamers and basically not transformed to the closed state (Supplementary Movie 9). In fact, they only rarely altered to closed rings during HS-AFM imaging

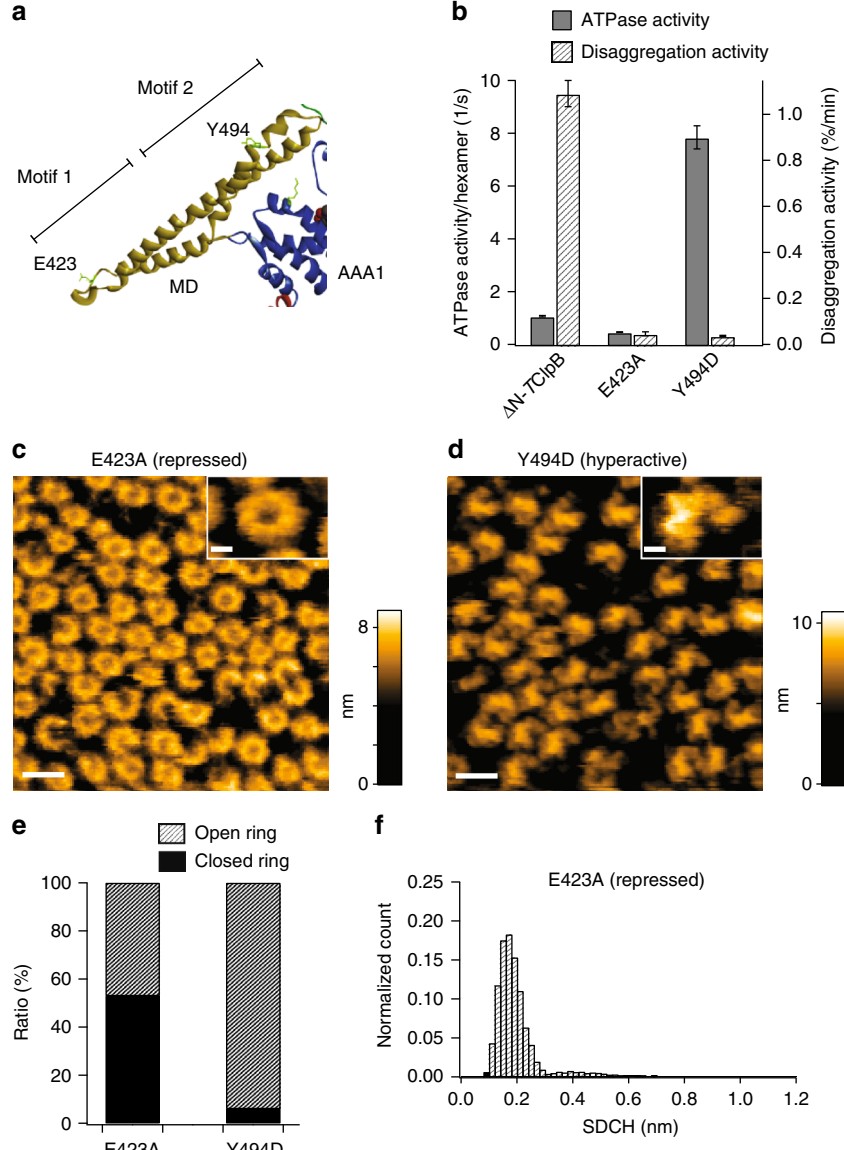

**Fig. 4** Oligomeric forms and ATPase and disaggregation activities of activity mutants. **a** The structure of MD connecting with AAA1 of *T*ClpB. The positions of residues replaced for repressed (E423A) and hyperactive (Y494D) mutants are indicated. **b** ATPase and disaggregation activities of ΔN-*T*ClpB, repressed E423A and hyperactive Y494D mutants. Data are means ± s.d. **c, d** Typical HS-AFM images of **c** E423A and **d** Y494D mutants. Scale bars, 20 nm. Z color bar, **c** 0 to 8.9 nm and **d** 0 to 10.8 nm. Insets at the upper right show magnified images of respective typical oligomeric forms. Scale bars, 5 nm. **e** Percentages of closed (black area) and open (hatched area) forms of E423A (472 molecules) and Y494D (378 molecules). **f** Histogram of SDCH along hexameric rings of E423A (4810 images from 9 molecules). The major distribution is around 0.2 nm, indicating that most of the closed rings adopt round forms

even at high [ATP] (Supplementary Figure 16c). This result strongly suggests that the open-ring forms of Y494D mutant (and also possibly open-ring forms of ΔN-*T*ClpB) are responsible for the high intrinsic disaggregation activity.

## Discussion

HS-AFM images of *T*ClpB in the presence of ATP demonstrated that *T*ClpB oligomers are highly flexible and undergo large-scale transitions among distinct oligomeric states. Roughly, the oligomers are classified into open and closed rings. The fraction of closed rings increases with increasing [ATP], of which dependence is similar to that of ATP binding to *T*ClpB, indicating that ATP binding induces an open-to-close transition during the ATPase cycle. However, the fraction of closed rings is saturated

around 60%. Even with a high concentration of ATPγS, it is saturated at ~70%. These results suggest that, both open and closed rings are in equilibrium even when all nucleotide binding sites are occupied by ATP or ATPγS.

In the presence of ATP, the closed rings adopt and transit the round, spiral and twisted-half-spiral forms, as well as intermediates among these three forms. The twisted-half-spiral form was newly found in the present study for *T*ClpB. This form was observed to occur more frequently than the spiral form, especially in low [ATP] (see Supplementary Movies 1 and 2). The frequency of transition cycles between round and distorted closed rings as assessed by SDCH increased with increasing [ATP] and reached a saturated value around 0.6 s$^{-1}$ at high [ATP], almost half of the saturated value of ATPase activity per hexamer (~1 s$^{-1}$). This

result indicates that the hydrolysis of 1–2 ATP molecules per hexamer induces a cycle of the structural change. Furthermore, the frequency of round ring appearance increased in high [ATP] or in 1 mM ATPγS, whereas ADP deforms the rings to spiral, twisted-half-spiral and even open forms (Fig. 1i and Supplementary Figure 7b). These results strongly suggest that the round-to-distorted state transitions occur in the step of ATP hydrolysis.

It should be noted that the appearance frequency of the closed ring and the frequency of transition between round and distorted closed rings were highly dependent on ATP concentration. Furthermore, when ATP was added during HS-AFM imaging of ΔN-TClpB without ATP preincubation, we observed gradual formation of oligomer rings (Supplementary Figure 13). These results indicate that the effects of surface-sample and tip-surface interactions on the observed dynamics are negligible.

HS-AFM imaging of the Walker mutants dissected the distinct roles of the AAA1 and AAA2 modules in the formation and structural dynamics of TClpB oligomers (Fig. 5). ATP binding to AAA1 is essential for the ring formation, consistent with previous studies[14,15,19,46], while ATP binding to AAA2 is essential for the formation of hexameric rings. ATP hydrolysis in AAA2 is crucial for driving large-scale structural changes in the hexameric rings, and ATP hydrolysis in AAA1 possibly contributes to increasing the coupling efficiency of these chemical and structural events. Although the conformational dynamics of Hsp104 oligomers has been considered to be triggered by ATP hydrolysis in AAA1[17], this inconsistency possibly stems from switched roles of AAA1 and AAA2 in Hsp104 and ClpB[14,15,19,37,47,48].

HS-AFM observations of activity mutants clearly demonstrated that conformational dynamics of TClpB is linked to the disaggregation activity. A substantial conformational difference in the MD between repressed and hyperactive mutants of EClpB has been found by cryo-EM analysis, revealing that MDs surrounding the core of hexameric ring are horizontally oriented in the repressed mutant, whereas they are rather vertically tilted in the hyperactive mutant[28]. In the horizontally oriented MDs, inter-MD connections are formed between motif 1 and motif 2 from adjacent MDs, resulting in tightening of the hexameric ring. In contrast, the inter-MD contacts are broken in the vertically tilted MDs. This MD-controlled structural flexibility change in the

hexameric ring has been also reported for Hsp104 in a cross-linking coupled mass-spectrometry study[49]. This variable MD orientation model can explain our HS-AFM images showing that the repressed mutant mostly adopts a round closed-ring state, whereas the hyperactive mutant adopts an open-ring state (Fig. 4). According to the structure of E423A mutant imaged by HS-AFM, the round closed ring would correspond to the repressed structure in which the inter-MD interactions are tightened. This suggests that the inter-MD interactions stabilize the round closed-ring and the disruption of these interactions is needed for the round-to-distorted structural transitions (Fig. 5). In the repressed E423A mutant, by the stabilized inter-MD connections, the structural transition to the distorted closed ring is probably inhibited even when ATP is hydrolyzed in AAA2. Alternatively, ATP binding and hydrolysis in AAA2 may allosterically regulate the orientation of MDs (Fig. 5), and hence E423A mutation may decouple the allosteric regulation. On the other hand, the open rings of hyperactive Y494D mutant are expected to show dynamic conformational changes because of weak interactions among protomers. Nevertheless, in its HS-AFM observation, they only rarely showed a transition to closed rings. Interestingly, most of the closed rings of Y494D mutant were in the round ring state (Supplementary Figure 16c). This result implies that in the Y494D mutant, the distorted closed rings are unstable and rapidly change to the open rings, which would not be able to be captured clearly in our HS-AFM observations. Structural instability of the distorted closed ring possibly accelerates the ATPase cycle of Y494D mutant, resulting in a high ATP hydrolysis activity (~8 s$^{-1}$).

The structural transitions of the hexameric TClpB ring observed with HS-AFM are basically consistent with the ratchet-like polypeptide translocation mechanism recently proposed for the disaggregation function of Hsp104[17]. In this model, the position of open site in a hexamer ring corresponding to a seam of the spiral shifts in one direction, resulting in processive rotary translocation. This model is attractive and seems plausible, considering the rotational propagation of chemical and structural states in rotor-less F$_1$-ATPase[41,50]. On the other hand, our HS-AFM movies did not show clear rotary shift of a seam or an open site position along the ring. Rather, two seams often appear

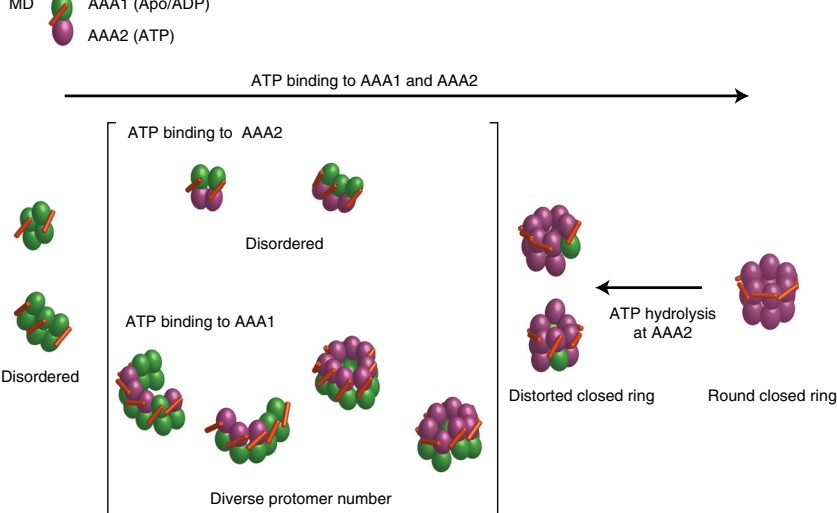

**Fig. 5** Schematic model of oligomeric structures and structural dynamics of ClpB. Green and purple colors of the AAA modules indicate the ADP/Apo state and the ATP bound state, respectively. ATP binding to AAA1 is essential for assembly into oligomers but insufficient to form ring-shaped hexamers. ATP binding to AAA2 regulates and adjusts the number of subunits in an oligomer to six (hexamer), and arranges MD in a horizontal configuration, resulting in round closed hexamers. ATP hydrolysis at AAA2 disrupts the inter-subunit interaction to form distorted closed hexamers in which some MDs are tilted. ATP hydrolysis on AAA1 possibly contributes to increasing the coupling efficiency of ATP hydrolysis on AAA2 and the structural transition

simultaneously at opposite sites in the ring, giving the twisted-half-spiral form. This might imply that ratchet-like substrate translocation is exerted randomly rather than sequentially among the subunits of the ClpB ring[43]. Furthermore, transitions between open and closed rings may be required for threading of a single polypeptide chain into the pore of the ClpB ring. To verify the mechanism further, HS-AFM imaging of the ClpB ring in the presence of substrate protein will be required.

## Methods

**Sample preparation.** Site-directed mutagenesis was performed by overlap extension PCR[51,52] by using the recombinant plasmid pMCB1[7] containing a ClpB gene from *Thermus thermophilus* as a template (Supplementary Table 2). The DNA sequences of the resulting expression plasmids were confirmed by DNA sequencing. Rabbit pyruvate kinase and chicken lactate dehydrogenase were purchased from Oriental Yeast. Enhanced yellow fluorescent protein (EYFP), *T*DnaK, *T*DnaJ, *T*GrpE, *T*ClpB, and its mutants were expressed in *E. coli* BL21(DE3) or KRX carrying pET23a-EYFP[34], pMDK6[53], pMDJ10[53], pMGE3[54], or pMCB1 with or without mutations, and purified as described previously[29,34,55,56]. For biotinylation of ΔN-*T*ClpB-Q142C, 20 μM ΔN-*T*ClpB-Q142C in a buffer (50 mM MOPS-NaOH pH7.0, 150 mM KCl, 5 mM MgCl$_2$) was incubated with 100 μM biotin-PEAC$_5$-maleimide (Dojin) at 37 °C for 1.5 h. The reaction mixture was applied to a NAP5 desalting column (GE Healthcare) equilibrated with a buffer (50 mM MOPS-NaOH pH7.5, 150 mM KCl, 5 mM MgCl$_2$) to remove the unreacted biotin-PEAC$_5$-maleimide.

**HS-AFM imaging.** HS-AFM experiments were performed using a laboratory-built instrument[57]. A freshly cleaved mica surface was used as a solid substrate. Before HS-AFM imaging, *T*ClpB samples in a buffer (50 mM MOPS pH 7.5, 150 mM KCl, 5 mM MgCl$_2$) was incubated at 55 °C in the presence of 2 mM ATP for 1 min. Then, a sample droplet of approximately 2 μl was placed on the mica substrate. After incubation for 3 min at 25 °C, the mica was thoroughly washed by a buffer (50 mM MOPS pH 7.5, 20 mM KCl, 5 mM MgCl$_2$) with or without nucleotides to remove excess molecules and change the buffer condition for HS-AFM imaging. Then, the sample stage was immersed in a liquid pool with 70 μl of the imaging buffer. The salt concentration was a key to successful imaging of *T*ClpB because high salt concentrations such as 150 mM KCl weakened the affinity of molecules to mica substrate, resulting in fast diffusion of molecules and thus hampering imaging. Therefore, we used a lower concentration of KCl (20 mM) which enabled moderate binding of ClpB onto mica substrate.

HS-AFM was operated in tapping mode using a small cantilever with a resonant frequency of ~0.8 MHz, a quality factor of ~2, and a spring constant of ~0.2 N m$^{-1}$. The cantilever's free oscillation amplitude was set at 1–2 nm and the set-point amplitude for feedback control was ~90% of the free oscillation amplitude.

**Analysis of HS-AFM image.** All HS-AFM images were processed by a low-pass filter with different cut-off frequencies (Fig. 1a, 2.0 nm, Fig. 1b–e, 4.0 nm, Fig. 2a, 3.7 nm, Fig. 3a, 2.0 nm, Fig. 3d, 3.4 nm, Fig. 4c, 2.5 nm, Fig. 4d, 3.5 nm) to suppress spike noises and enhance image features. The 1st order plane flatten filter was applied to compensate for tilt of the sample relative to the scanning plane.

To estimate the circularity of oligomer shapes, we first carried out binary digitization for each AFM image of a *T*ClpB oligomer by setting an optimum threshold to obtain the X and Y coordinates of a two-dimensional (2D) outline of the oligomer. The outline was processed by elliptic Fourier transformation[58] and reconstructed by Fourier coefficients up to the 8th order, resulting in a smoothed profile (Supplementary Figure 7a). The circularity of the profile is defined by $4\pi$S L$^{-2}$, where L and S are its perimeter and the area surrounded by the profile, respectively (Supplementary Figure 7a). After making histograms of circularity, the threshold to distinguish between the closed and open rings was determined from the overall distributions. In this study, we used 0.68 for the threshold, judging from histograms shown in Fig. 1h.

To evaluate corrugations of the cross-sectional top surface profile of *T*ClpB oligomers and discriminate between round and distorted closed rings, we used the standard deviation of cross-sectional height variations (SDCH). We first calculated the center of a ring by using its outline described above. Then the outline of the ring was shrunk towards the center by a fraction of 0.6–0.7. The height profile along the shrunk outline was used to calculate the value of SDCH. When an image rarely showed large spike noises, we did not count its produced large SDCH value and instead we assumed the same SDCH as that of the previous frame image. The threshold to distinguish between the round and distorted closed rings was set at 0.3 nm from the overall histogram distributions shown in Fig. 2c. All image analyses and AFM-image simulations were carried out using a software program that we developed based on IgorPro (WaveMetrics, Inc., Lake Oswego, OR).

**Analysis of ATP binding to ΔN-*T*ClpB.** Mant-ADP, a fluorescent nucleotide analogue, was purchased from Thermo Fisher Scientific. Purified ΔN-*T*ClpB (1 μM as hexamer) in a buffer (50 mM MOPS pH 7.5, 20 mM KCl, 5 mM MgCl$_2$) was

incubated at 25 °C, and Mant-ADP was added at concentrations indicated in Supplementary Figure 8. After the 1-min incubation, their fluorescence spectra were measured over 400–500 nm with excitation at 360 nm. As a control, the fluorescence spectra of Mant-ADP solutions alone with concentrations identical to those used above were also measured in the same buffer. The differences of the emission intensity at 440 nm with and without ΔN-*T*ClpB was plotted against the Mant-ADP concentration. Apparent dissociation constants of Mant-ADP for ΔN-*T*ClpB (as monomer) was calculated by fitting the titration data to the model of simple 1:1 binding in consideration of the decrease of concentrations of free Mant-ADP[15]. The replacement of Mant-ADP by ATP was detected by monitoring the decrease in fluorescence at 440 nm by adding Mg-ATP at various concentrations to the mixture of ΔN-*T*ClpB (1 μM as hexamer) and 12.5 μM Mant-ADP. Dissociation constant of ATP for ΔN-*T*ClpB (as monomer) were calculated by fitting the titration data to the competitive binding model[15]. All measurements were performed using an FP-8500 spectrofluorometer (JASCO).

**Analysis of ATPase activity.** ATPase activities of ΔN-*T*ClpB mutants were measured spectrophotometrically using an ATP-regenerating system at 25 °C. The reaction mixture consists of 50 mM MOPS-NaOH (pH 7.5), 20 mM KCl, 5 mM MgCl$_2$, 2.5 mM phosphoenolpyruvate, 0.2 mM NADH, 50 μg ml$^{-1}$ pyruvate kinase, 50 μg ml$^{-1}$ lactate dehydrogenase, and indicated concentrations of ATP. ΔN-*T*ClpB mutants (final concentration was 0.25 μM as hexamer) were added to the reaction mixture and the absorbance change at 340 nm was monitored for each sample using a V-650 spectrophotometer (JASCO).

**Analysis of disaggregation activity.** EYFP (12.0 μM monomer) in a mixture containing 50 mM MOPS-NaOH (pH 7.5), 150 mM KCl, 10 mM MgCl$_2$, and 5 mM tris(2-carboxyethyl)phosphine (TCEP) was aggregated by incubation at 80 °C for 10 min. The aggregated EYFP was diluted 40-fold into a buffer (50 mM MOPS-NaOH pH 7.5, 20 mM KCl, 10 mM MgCl$_2$, and 5 mM ATP) and the monitoring of EYFP fluorescence at 527 nm with excitation at 513 nm was initiated. After 2-min incubation at 55 °C, *T*DnaK (0.6 μM monomer), *T*DnaJ (0.2 μM monomer), *T*GrpE (0.1 μM dimer), and ΔN-*T*ClpB or its mutants (0.05 μM as hexamer) were added to the mixture and the incubation was continued. The disaggregation rates were estimated as follows. Slopes of a fluorescence change were calculated by using any consecutive 1-min data in the initial 5-min measurement after starting the reaction. The slope showing the maximum value was used as the disaggregation rate. Fluorescence measurements were performed using an FP-8500 spectrofluorometer.

**Negative-staining electron microscopy.** For negative-staining electron microscopy (EM), the samples of full-length *T*ClpB (1.36 μM as hexamer) and ΔN-*T*ClpB (0.92 μM as hexamer) suspended in a buffer (50 mM MOPS-KOH pH7.5, 20 mM KCl, 5 mM MgCl$_2$) were incubated with 2 mM ATP for 1 min at 55 °C. For EM observation, the concentrations of these proteins were adjusted with the same solution. The specimen suspensions (2.5 μl) were applied onto a carbon-coated copper grid that had been glow-discharged beforehand. After removing the excess sample solution with a filter paper, the specimens were stained with a 2% (w v$^{-1}$) uranyl acetate solution for 30 s. The grids were dried in air after removing the staining solution with a filter paper. EM images were recorded on DE-20 camera (Direct electron LP) at a nominal magnification of 30,000 using a JEM2200FS electron microscope (JEOL Ltd.) operated at 200 kV accelerating voltage. The energy slit width was adjusted at 20 eV. The image pixel size was 2 Å on the camera. A 2–3 μm under-focused condition was selected to enhance the contrast of EM images. The image analysis was performed with Relion 2.0[59] after subjecting the images to motion correction using the DE_process_frames.py script provided by the manufacturer. The particles were automatically picked up from the images after individually correcting the contrast transfer function. The particles of full-length *T*ClpB and ΔN-*T*ClpB were classified as two-dimensional (2D) particles. Imposing symmetry and calculation of dot products in the 2D class averages were performed with EMAN 1.9 software[60].

**Sedimentation velocity analytical ultracentrifugation.** Sedimentation velocity analytical ultracentrifugation (SV-AUC) experiments were performed at 20 °C using a ProteomeLab XL-I analytical ultracentrifuge (Beckman Coulter) equipped with Rayleigh interference optics, at 42,000 rpm. The sedimentation coefficient distributions were obtained using the c(s) method of SEDFIT ver. 15.01b[61]. The frictional ratio, meniscus, and time-invariant noise were floated during the fitting procedure, and a regularization level of 0.68 was used. The partial specific volumes of full-length *T*ClpB and ΔN-*T*ClpB were calculated to be 0.7452 cm$^3$ g$^{-1}$ and 0.7444 cm$^3$ g$^{-1}$, respectively, using the program SEDNTERP 1.09. The buffer density and viscosity of each condition were determined using DMA 5000 densitometer (Anton Paar) and Lovis 2000M viscometer (Anton Paar), respectively. To assess the effect of protein concentration on the oligomeric state of full-length *T*ClpB and ΔN-*T*ClpB, experiments were conducted at 0.5 mg m L$^{-1}$ (0.9 μM as hexamer for full-length *T*ClpB and 1.0 μM as hexamer for ΔN-*T*ClpB) and 2.0 mg m L$^{-1}$ (3.6 μM as hexamer for full-length *T*ClpB and 4.0 μM as hexamer for ΔN-*T*ClpB) in the presence of 20 mM ATP or ADP, or 2 mM ATPγS. The protein samples in a buffer (50 mM MOPS pH 7.5, 50 mM KCl, 5 mM MgCl$_2$) were

preincubated at 55 °C in the presence of 20 mM ATP, 2 mM ATPγS or 20 mM ADP for 1 min, before their application to SV-AUC.

**Native mass spectrometry**. Native mass spectrometry (nMS) was performed according to the method used in the study by Ishii et al with a minor modification[62]. The samples of full-length $T$ClpB (13 μM as hexamer) and ΔN-$T$ClpB (9 μM as hexamer) in a buffer (50 mM MOPS pH 7.5, 150 mM KCl, 5 mM MgCl$_2$) were incubated at 55 °C in the presence of 2 mM ATP or ATPγS for 1 min. Then, the samples were buffer-exchanged into 150 mM ammonium acetate buffer, pH 7.5, containing 100 μM ATP or ATPγS, at 4 °C by passing the samples through MicroBioSpin-6 columns (Bio-Rad, Hercules, California, USA). The buffer-exchanged samples were immediately analyzed by nanoflow electrospray ionization mass spectrometry using gold-coated glass capillaries made in house (~2–5 μl sample loaded per analysis). Spectra were recorded on a SYNAPT G2-S$i$ HDMS mass spectrometer (Waters, Massachusetts, Milford, USA) in positive ionization mode at 1.33 kV with a 150 V sampling cone voltage and source offset voltage, 0 V trap and transfer collision energy, and 5 m min$^{-1}$ trap gas flow. The spectra were calibrated using 1 mg/ml cesium iodide and analyzed using MassLynx software (Waters).

**Data availability**. Data supporting the findings of this manuscript are available from the corresponding authors upon reasonable request. The raw AFM image files, data of protein biochemical assessments, disaggregation activities, the raw cryo-EM image files, and the data sets of SV-AUC and nMS are available from the authors on request. Source codes for AFM image analysis are also available from the authors on request. The nMS proteomics data associated with this manuscript have been deposited to the ProteomeXchange Consortium via the PRIDE partner repository with the dataset identifier PXD009732.

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

## Acknowledgements

This work was supported in part by JSPS KAKENHI, JP15H03540 (to T.U.), JP17H03975 (to S.U.), JP26440085 (to Y.W.), JP15H04366 and JP17K19213 (to R.I.) and JP24227005 (to T.A.), MEXT KAKENHI, JP16H00830 and JP16H00758 (to T.U.), JP16H00770 (to S. U.), JP16H00789, JP16H00858 (to R.I.) and JP26119003 (to T.A.), the Okazaki BIO-NEXT project of National Institutes of Natural Sciences (to S.U.) and JST/CREST, JPMJCR13M1 (to T.A.).

## Author contributions

T.U., Y.W., R.I. and T.A. conceived and designed research; Y.W., Y.N. and T.Y. prepared sample and performed biochemical analysis. T.U. and T.A. set up HS-AFM imaging system including both hardware and software. T.U. and H.W. performed HS-AFM imaging and analysed the data. T.M. and S.U performed sedimentation velocity analytical ultracentrifugation. K.I. and S.U. performed native mass spectrometry. C.S. and K.M. performed negative-staining electron microscopy. T.U., Y.W., S.U., K.M., R.I. and T.A. wrote the paper.

## Additional information

**Competing interests:** The authors declare no competing interests.

