## [Peer Review File · Nature Communications]

Reviewers' comments:

Reviewer #1 (Remarks to the Author):

Uchihashi et al present interesting and novel results on a AAA+ disaggregase using high speed AFM, giving some new insights into the roles of ATP binding and hydrolysis, distinct contributions of AAA1 and AAA2, and middle domain regulation. The presented AFM images are of good quality, with the static pictures highlighting differences between the different mutants, in particular Fig. 3. The AFM movies complement these static records with information about ClpB dynamics, although a more concise presentation might be more effective.

There are limitations to the extent of molecular interpretation possible with AFM, and the authors have gone slightly beyond what can be reliably determined from these images. The attribution of symmetry to the closed rings is not supported by recently published structures (ref 12 cited in support of this imposed the symmetry), and the AFM in plane resolution is not sufficient to base an interpretation of ring symmetry or the seam. They should use more accurate terminology - a closed ring is not necessarily symmetric. Similarly, it is hard to visualise what is meant by "twisted-half-spiral form". A cartoon would help, especially since this is presented as a possible novel conformation for ClpB. The paper would benefit from the addition of negative stain EM images of the various states being proposed on the basis of the AFM.

Can the authors comment on how the binding to the mica substrate may have affected their findings? e.g., to what extent might the dynamics be caused by the interaction with the AFM probe?

Can they provide additional evidence for the presumed ClpB binding orientation (N-terminal pointing upwards) on mica? Fig. S1a (full length TClpB) looks like a lower-quality version of Fig. 1 (NT deletion mutant), and not like "incomplete rings or circular particles with a high peak at their center" (first paragraph of Results section): This could mean that the NT (facing the AFM tip) obscures a clear view of underlying ring structure, but also that it (when the NT is facing the mica) affects the binding to the substrate, thus reducing the stability for AFM imaging.

The claim that deletion of the N-terminal domain has no impact on the ClpB chaperone activity is not entirely justified. There are some effects of the deletion and they have been documented in the literature - even in the titles of references 11 and 13 cited in support of the opposite claim. It would be more accurate to say that the deletion does not have a major impact.

A sequential rotary mechanism is dismissed with insufficient justification, considering that no observations are made in free solution. The AFM resolution is not sufficient to reliably identify the seam position in the closed ring. They can discern the open spiral with AAA1-AAA2 contacts quite well but rotary movement on a closed ring is more subtle and a sequential mechanism cannot be excluded on the basis of this work.

Minor comments:

The colour scale bar (shown in Fig. S1) should be included in the main Fig. 1, and its range specified for all AFM images in the manuscript.

SI, HS-AFM image analysis: The image processing procedures (averaging and flatten filter) should be specified more accurately. What type of average filter was used, over how many pixels did it average, and what physical distances does that correspond to in the various images? Was the flatten filter 0th, 1st, 2nd, ... order and applied line by line?

Reviewer #2 (Remarks to the Author):

In this study Uchihashi et al., use high speed AFM to investigate the structure and dynamic conformations of the ClpB AAA+ hexameric disaggregase. With HS-AFM the authors are able to show various ring-shaped hexamers and compare conformations under different nucleotide conditions and with ATPase mutants. They identify three general conformations of the hexamer: a symmetric ring, a two-turn spiral, and a pair of twisted half spirals. From their work they propose how these oligomeric forms could function in the ATPase cycle. While the HS-AFM method is quite powerful in resolving views of the ClpB hexamer, the conclusions about these three states seem substantially over-interpreted given the low-resolution of the images. Furthermore, much of the work presented involves comparison of open vs. closed-ring conformations, however it is unclear whether the open ring is a bonafide conformation or a partially dissociated complex containing 5 subunits or other arrangement. Finally, in light of recent structures of substrate-bound ClpB, Hsp104 and other AAA+ complexes that describe specific conformations at high-resolution, I feel this study does not provide substantial mechanistic or functional insight to warrant publication in Nature Communications. Specific concerns are described below.

-With these low-resolution views it is unlikely that the authors can conclude specifics about the conformational states including in particular the symmetry (or asymmetry) of the hexamer or whether the open and closed ring states contain six subunits. Additionally, the rings may remain asymmetric but the asymmetry may not be resolvable with HS-AFM. All recent structures of AAA+ complexes have described an asymmetric hexamer arrangement. From these 2d images one cannot conclude that the hexamer adopts a two-turn spiral or a pair of twisted half spirals – these conclusions seem to be interpreted based on previous structural studies (Fig 1f) but are inconclusive given the data presented here.

-In particular, the open-ring seems questionable given that it has not been described in other structural studies such as in cryo-EM studies in which many thousands of single particles are analyzed. Thus, one wonders whether these states are more artefactual and result from lack of an NTD, low nucleotide occupancy or other experimental conditions such as the use of a solid mica support or buffer washing steps. To address this, additional supportive experiments are important, such as size exclusion chromatography analysis, aUC, SAXS or cryoEM to support the ring-open state.

-In this study the authors use a ClpB construct without an N-terminal domain. The absence of this domain could affect the stability or conformation dynamics of the hexamer and thus these data may not reflect the full-length wild type complex.

Reviewer #3 (Remarks to the Author):

This study represents a very high technical standard and the results are convincing and impressive. Even the data analysis is very creative, as it is not easy to transfer many movies into quantitative measures of data sets to finally allow for the conclusions that have been convincingly drawn. There is just a couple of minor points:

What is the justification for setting the SDCH threshold to 0.3 nm?

The method for measuring the ATPase activity should be included into the main manuscript.

Fig 3 d and e: Why not putting data of walker a mutant here?

Fig 4 f: Data of the hyperactive mutant?

Response to the Reviewers' comments

The Reviewers' comments are shown in *Italic*.

First of all, we sincerely appreciate the valuable comments raised by the Reviewers. According to the comments, we have added not only the new data of high-speed AFM (HS-AFM), but also the data of negative-staining electron microscopy (EM), sedimentation velocity analytical ultracentrifugation (SV-AUC), native mass spectrometry (nMS). We have added new coauthors who conducted additional experiments, and revised the manuscript extensively. All changes are highlighted with red color in the revised manuscript. With the help of the Reviewers, we believe that the revised manuscript is significantly improved and suitable for publication in the Nature Communications.

Reviewer #1 (Remarks to the Author):

Uchihashi et al present interesting and novel results on a AAA+ disaggregase using high speed AFM, giving some new insights into the roles of ATP binding and hydrolysis, distinct contributions of AAA1 and AAA2, and middle domain regulation. The presented AFM images are of good quality, with the static pictures highlighting differences between the different mutants, in particular Fig. 3. The AFM movies complement these static records with information about ClpB dynamics, although a more concise presentation might be more effective.

Response: We appreciate positive evaluation by the Reviewer #1.

There are limitations to the extent of molecular interpretation possible with AFM, and the authors have gone slightly beyond what can be reliably determined from these images. The attribution of symmetry to the closed rings is not supported by recently published structures (ref 12 cited in support of this imposed the symmetry), and the AFM in plane resolution is not sufficient to base an interpretation of ring symmetry or the seam. They should use more accurate

terminology - a closed ring is not necessarily symmetric. Similarly, it is hard to visualise what is meant by “twisted-half-spiral form”. A cartoon would help, especially since this is presented as a possible novel conformation for ClpB. The paper would benefit from the addition of negative stain EM images of the various states being proposed on the basis of the AFM.

Response: We appreciate important comment on terminology. We have revised the all phrases of “symmetric ring” to “pseudo-symmetric ring”, and added following description: “One form is a pseudo-symmetric hexameric ring, although the resolution of HS-AFM lower than X-ray crystallography and cryo-EM single-particle analysis prevented us from judging if the ring is really symmetric (**Fig. 1b** and **Supplementary Fig. 3a**).” (page 6, line 11 from top). About the twisted-half-spiral form, we have added description about one possible structural model as follows: “One possible structural model of this form is that the interactions between the protomer 1 and 6, and 3 and 4 in **Fig. 1f** are disrupted and then relative height of the two trimer units (P1-3 and P4-6) is shifted (**Supplementary Fig. 3e**).” (page 7, line 2 from top). We have also added negative-staining EM images of full-length and ΔN -TClpB molecules showing the closed and open hexamer rings in the presence of ATP (**Supplementary Fig. 4 and 5**), and added following description: “Similar to the HS-AFM observation, both open and closed rings were observed with negative-staining electron microscopy for both the full-length and ΔN -TClpB (**Supplementary Fig. 4 and 5**). These results indicate that the open form is an inherent structure of the full-length ClpB and not an artifact caused by deletion of the N-terminal domain.” (page 7, line 9 from top).

Can the authors comment on how the binding to the mica substrate may have affected their findings? e.g., to what extent might the dynamics be caused by the interaction with the AFM probe?

Response: We have added following description in the Discussion section of the revised manuscript: “It should be noted that the ratio of the closed ring and the

frequency of transition between pseudo-symmetric and asymmetric rings were highly dependent on the nucleotide conditions such as different ATP concentrations and usage of ATP γ S. Furthermore, when ATP was added during HS-AFM imaging of Δ N-TClpB without ATP preincubation, we observed gradual formation of oligomer rings (**Supplementary Fig. 13**). These results indicate that the effects of the interaction with the solid substrate or the external force from the AFM tip on the observed dynamics were negligible.” (page 18, line 3 from top).

Can they provide additional evidence for the presumed ClpB binding orientation (N-terminal pointing upwards) on mica? Fig. S1a (full length TClpB) looks like a lower-quality version of Fig. 1 (NT deletion mutant), and not like “incomplete rings or circular particles with a high peak at their center” (first paragraph of Results section): This could mean that the NT (facing the AFM tip) obscures a clear view of underlying ring structure, but also that it (when the NT is facing the mica) affects the binding to the substrate, thus reducing the stability for AFM imaging.

Response: To show the shapes of full-length TClpB more clearly, we replaced the image of the **Supplementary Fig. 1a**. To show the additional evidence of TClpB binding orientation, we observed binding of streptavidin to the biotinylated Δ N-TClpB (Q142C-biotin) on the mica substrate, and added following description: “The orientation was further confirmed by binding of streptavidin to the top surface of Δ N-TClpB biotinylated at the N-terminal side (Δ N-TClpB-Q142C-biotin) (**Supplementary Fig. 1b-d**).” (page 6, line 4 from top).

The claim that deletion of the N-terminal domain has no impact on the ClpB chaperone activity is not entirely justified. There are some effects of the deletion and they have been documented in the literature - even in the titles of references 11 and 13 cited in support of the opposite claim. It would be more accurate to say that the deletion does not have a major impact.

Response: We compared disaggregation activities of full-length and Δ N-TCIpB and showed that the N-terminal deletion does not give a major impact on disaggregation activity of TCIpB (**Supplementary Fig. 2**). Based on this result, we revised the manuscript as follows: “Since the N-terminal deletion does not have a major impact on the chaperone activity of TCIpB (**Supplementary Fig. 2**), we hereafter use the Δ N-TCIpB as wild-type.” (page 6, line 6 from top).

A sequential rotary mechanism is dismissed with insufficient justification, considering that no observations are made in free solution. The AFM resolution is not sufficient to reliably identify the seam position in the closed ring. They can discern the open spiral with AAA1-AAA2 contacts quite well but rotary movement on a closed ring is more subtle and a sequential mechanism cannot be excluded on the basis of this work.

Response: We agree with the comment. We have revised and shortened the discussion about the mechanism as follows: “This might imply that ratchet-like substrate translocation is exerted by a mechanism of “hand-over-hand hauling” rather than a sequential rotary mechanism. Alternatively, interaction with the substrate peptide chain is required to exert the sequential rotary mechanism. To verify the mechanism further, HS-AFM imaging of the ClpB ring in the presence of the substrate chain will be required.” (page 20, line 12 from top).

Minor comments:

The colour scale bar (shown in Fig. S1) should be included in the main Fig. 1, and its range specified for all AFM images in the manuscript.

Response: The colour scale bars have been added to all AFM images.

SI, HS-AFM image analysis: The image processing procedures (averaging and flatten filter) should be specified more accurately. What type of average filter was

used, over how many pixels did it average, and what physical distances does that correspond to in the various images? Was the flatten filter 0th, 1st, 2nd, ... order and applied line by line?

Response: All HS-AFM images were processed by a low-pass filter with different cut-off frequencies (Fig. 1a, 2.0 nm, Fig. 1b-e, 4.0 nm, Fig. 2a, 3.7 nm, Fig. 3a, 2.0 nm, Fig. 3d, 3.4 nm, Fig. 4c, 2.5 nm, Fig. 4d, 3.5 nm) to suppress spike noises and enhance image features. The 1st order plane flatten filter was applied to compensate for tilt of the sample relative to the scanning plane. The detail of image processing procedures has been added in the Methods section of the revised Supplementary materials (page 3, line 6 from bottom).

Reviewer #2 (Remarks to the Author):

In this study Uchihashi et al., use high speed AFM to investigate the structure and dynamic conformations of the ClpB AAA+ hexameric disaggregase. With HS-AFM the authors are able to show various ring-shaped hexamers and compare conformations under different nucleotide conditions and with ATPase mutants. They identify three general conformations of the hexamer: a symmetric ring, a two-turn spiral, and a pair of twisted half spirals. From their work they propose how these oligomeric forms could function in the ATPase cycle. While the HS-AFM method is quite powerful in resolving views of the ClpB hexamer, the conclusions about these three states seem substantially over-interpreted given the low-resolution of the images. Furthermore, much of the work presented involves comparison of open vs. closed-ring conformations, however it is unclear whether the open ring is a bonafide conformation or a partially dissociated complex containing 5 subunits or other arrangement. Finally, in light of recent structures of substrate-bound ClpB, Hsp104 and other AAA+ complexes that describe specific conformations at high-resolution, I feel this study does not provide substantial mechanistic or functional insight to warrant publication in Nature Communications. Specific concerns are described below.

Response: We appreciate the comment by the Reviewer #2, although we would like to decline to agree fully. To address the comment raised by the Reviewer #2, we carried out negative-staining EM observation of full-length and ΔN -TClpB in the presence of ATP (**Supplementary Fig. 4 and 5**). Both full-length and ΔN -TClpB showed closed and open rings presumably consist of six subunits. This result indicates that the open ring is a bonafide conformation but not a partially dissociated complex containing 5 subunits or other arrangement. As the Reviewer #2 pointed out, excellent structural studies of ClpB and Hsp104 have been reported recently. We agree that recent cryo-EM single particle analysis provides valuable information about the functional structure of ClpB and Hsp104. However, most of works have been carried out in the presence of poorly-hydrolysable or non-hydrolysable ATP analogue such as ATP γ S or AMPPNP, but not ATP. Under these conditions, information on conformational dynamics cannot be obtained. On the other hand, our HS-AFM observation has directly visualized conformational dynamics of ClpB in the presence of ATP. In this context, we believe that our study provides substantial mechanistic or functional insight of ClpB.

-With these low-resolution views it is unlikely that the authors can conclude specifics about the conformational states including in particular the symmetry (or asymmetry) of the hexamer or whether the open and closed ring states contain six subunits. Additionally, the rings may remain asymmetric but the asymmetry may not be resolvable with HS-AFM. All recent structures of AAA+ complexes have described an asymmetric hexamer arrangement. From these 2d images one cannot conclude that the hexamer adopts a two-turn spiral or a pair of twisted half spirals – these conclusions seem to be interpreted based on previous structural studies (Fig 1f) but are inconclusive given the data presented here.

Response: We agree that the resolution of HS-AFM is not enough to resolve the symmetry of the ring. We have revised the all phrases of “symmetric ring” to

“pseudo-symmetric ring”, and added following description: “One form is a pseudo-symmetric hexameric ring, although the resolution of HS-AFM lower than X-ray crystallography and cryo-EM single-particle analysis prevented us from judging if the ring is really symmetric (**Fig. 1b** and **Supplementary Fig. 3a**).” (page 6, line 11 from top). On the other hand, we believe that our HS-AFM observations have successfully resolved that the hexamer adopts a two-turn spiral or a pair of twisted half spirals. In the SV-AUC of full-length and ΔN -TCIpB (**Supplementary Fig. 9**), we found that the trimer is a major component in the presence of ATP, and added following description: “To quantitatively analyze the number of protomers in an oligomer of full-length or ΔN -TCIpB in solution, we carried out sedimentation velocity analytical ultracentrifugation (SV-AUC) (**Supplementary Fig. 9**). Interestingly, we found that trimers are the major and hexamers and other oligomers are minor for both full-length and ΔN -TCIpB in the presence of 20 mM ATP (**Supplementary Fig. 9a to c**, red lines). This result is consistent with the twisted-half-spiral form that consists of two trimer units.” (page 8, line 5 from bottom).

-In particular, the open-ring seems questionable given that it has not been described in other structural studies such as in cryo-EM studies in which many thousands of single particles are analyzed. Thus, one wonders whether these states are more artefactual and result from lack of an NTD, low nucleotide occupancy or other experimental conditions such as the use of a solid mica support or buffer washing steps. To address this, additional supportive experiments are important, such as size exclusion chromatography analysis, aUC, SAXS or cryoEM to support the ring-open state.

Response: To address the comment raised by the Reviewer #2, we carried out negative-staining EM observation of full-length and ΔN -TCIpB in the presence of ATP (**Supplementary Figs. 4 and 5**, and description in page 7, line 9 from top). As a result, for both the full-length and ΔN -TCIpB, closed and open rings were observed. This result indicates that the open ring is not an artifact due to the lack of NTD. In addition to the open and closed rings, we have also observed smaller

oligomers in the EM images, consistent with the results of SV-AUC (**Supplementary Fig. 9**) and nMS (**Supplementary Fig. 10**). We think that previous EM works excluded the open rings and smaller oligomers from the single particle analysis, because the open ring form is a minor conformation under the ATP γ S condition used in the previous study, as shown in **Figs. 1i**. Furthermore, in SV-AUC and nMS, nucleotide-dependent oligomeric states of full-length and Δ N-TClpB were similar. Especially, in SV-AUC in the presence of ATP, trimer was the major component for both of full-length and Δ N-TClpB, consistent with our HS-AFM observation of the “twisted-half-spiral form”. These results support the notion that there are no major effects due to the lack of the NTD.

-In this study the authors use a ClpB construct without an N-terminal domain. The absence of this domain could affect the stability or conformation dynamics of the hexamer and thus these data may not reflect the full-length wild type complex.

Response: As described above, there are no essential differences between the oligomeric behaviors of full-length and Δ N-TClpB. Furthermore, we compared disaggregation activities of full-length and Δ N-TClpB and demonstrated that the lack of NTD does not give a major impact on disaggregation activity of TClpB (**Supplementary Fig. 2**). Because the NTD contributes to the substrate-binding of ClpB, deletion of the NTD can change its substrate-binding property, and in some case, decrease its disaggregation activity (Barnett, M.E. et al., J Biol Chem 280, 34940, 2005; Lee, S. et al., Mol Cell 25, 261, 2007; Mizuno, S. et al., FEBS J 279, 1474, 2012). However, under many experimental conditions, the NTD-truncated ClpB has been shown to have substantial disaggregation activity as in our current result (Barnett, M.E. et al., J Biol Chem 280, 34940, 2005; Mizuno, S. et al., FEBS J 279, 1474, 2012; Beinker, P. et al., J Biol Chem 277, 47160, 2002; Nagy, M. et al., J Mol Biol 396, 697, 2010). In addition, it has been reported that stability of oligomeric structure is hardly influenced by the NTD truncation (Beinker, P. et al., J Biol Chem 277, 47160, 2002; Barnett, M.E. et al.,

J Biol Chem 275, 37565, 2000). Furthermore, many ClpB genes, including the gene of TClpB, have two translation-initiation sites, and at least in *E. coli*, both the full-length and the NTD-truncated version of ClpB were known to be expressed as physiologically functional molecules (Park, S.K. et al., J Biol Chem 268, 20170, 1993; Chow, I.T. et al., FEBS Lett 579, 4235, 2005; Chow, I.T. et al., FEBS Lett 579, 4242, 2005). Thus, we believe that the conformational changes observed for the Δ N-TClpB reflect the essence of the disaggregation function of ClpB, regardless of the lack of NTD.

Reviewer #3 (Remarks to the Author):

This study represents a very high technical standard and the results are convincing and impressive. Even the data analysis is very creative, as it is not easy to transfer many movies into quantitative measures of data sets to finally allow for the conclusions that have been convincingly drawn. There is just a couple of minor points:

Response: We appreciate high evaluation by the Reviewer #3.

What is the justification for setting the SDCH threshold to 0.3 nm?

Response: After making the histograms of the SDCH, the threshold to distinguish the pseudo-symmetric and asymmetric rings was set at 0.3 nm from the overall distributions. This has been clarified in the Methods section of the revised Supplementary materials (page 4, line 3 from bottom).

The method for measuring the ATPase activity should be included into the main manuscript.

Response: We briefly described the method for ATPase activity measurement in the revised main manuscript (page 11, line 13 from top), in addition to the

Methods section of the Supplementary materials.

Fig 3 d and e: Why not putting data of walker a mutant here?

Response: We have added the sequential images and time course of SDCH of 2KT/AA mutant in the revised **Fig. 3d** and **e**, respectively. We did not add the data of 1KT/AA because this mutant did not form closed rings.

Fig 4 f: Data of the hyperactive mutant?

Response: SDHC plot of hyperactive Y494D mutant was difficult because this mutant almost showed open ring state.

Other points of revision

1. Takahiro Maruno, Kentaro Ishii, Susumu Uchiyama, Chihong Song, and Kazuyoshi Murata have been added as coauthors of the revised manuscript.
2. We have added the HS-AFM images (**Fig. S13**) and supplementary movies (**Movie S6**) showing formation of oligomer rings after the addition of ATP to ΔN -TCIpB without ATP preincubation during observation.
3. We have added the Data and code availability section in the revised manuscript (page 29).

Reviewers' comments:

Reviewer #1 (Remarks to the Author):

The authors have addressed almost all of the reviewer comments. I have 2 remaining issues, one minor terminology point and one substantial point of interpretation about the mechanism:

The terminology pseudo symmetric is unnecessarily confusing and misleading. They should just use "closed rings".

The response about a rotary mechanism is very puzzling:

"This might imply that ratchet-like substrate translocation is exerted by a mechanism of "hand-over-hand hauling" rather than a sequential rotary mechanism. Alternatively, interaction with the substrate peptide chain is required to exert the sequential rotary mechanism. "

This does not make sense. Hand over hand hauling is what has been suggested by the various crystal and single particle EM structures of AAA+ hexamers, including ClpB, and that would imply a rotary mechanism as the pore loops spiral around the translocating chain, not an alternative to such a mechanism.

Are the authors trying to suggest that the open rings and half twisted spiral transitioning to closed rings are somehow involved in the translocation?

They also say "Alternatively, interaction with the substrate peptide chain is required to exert the sequential rotary mechanism."

But without substrate, there is no translocation, so no rotary or other mechanism ?

Reviewer #3 (Remarks to the Author):

The paper has been significantly improved and is ready for publication.

Response to the Reviewers' comments

The Reviewers' comments are shown in *Italic*.

We sincerely appreciate the valuable comments raised by the Reviewers. According to the comments, we have revised manuscript. All changes are highlighted with red color in the revised manuscript. With the help of the Reviewers, we believe that the revised manuscript has been further improved and suitable for publication in the Nature Communications.

Reviewer #1 (Remarks to the Author):

The authors have addressed almost all of the reviewer comments. I have 2 remaining issues, one minor terminology point and one substantial point of interpretation about the mechanism:

The terminology pseudo symmetric is unnecessarily confusing and misleading. They should just use "closed rings".

Response: We appreciate the comment by the Reviewer #1. In our previous manuscript, we first classified ClpB oligomers into the "closed" and "open" rings, then, the closed rings were further classified into the "pseudo-symmetric" and "asymmetric" rings. Therefore, the "closed rings" has another meaning in the higher hierarchy in our manuscript. In the revised manuscript, we used the terminology "round closed ring" instead of "pseudo-symmetric ring", and "distorted closed ring" instead of "asymmetric ring".

The response about a rotary mechanism is very puzzling:

"This might imply that ratchet-like substrate translocation is exerted by a mechanism of "hand-over-hand hauling" rather than a sequential rotary mechanism. Alternatively, interaction with the substrate peptide chain is required

to exert the sequential rotary mechanism. “

This does not make sense. Hand over hand hauling is what has been suggested by the various crystal and single particle EM structures of AAA+ hexamers, including ClpB, and that would imply a rotary mechanism as the pore loops spiral around the translocating chain, not an alternative to such a mechanism.

Are the authors trying to suggest that the open rings and half twisted spiral transitioning to closed rings are somehow involved in the translocation?

They also say “Alternatively, interaction with the substrate peptide chain is required to exert the sequential rotary mechanism.”

But without substrate, there is no translocation, so no rotary or other mechanism ?

Response: We appreciate the comment by the Reviewer #1. We agree that rotary mechanism is not an alternative but also classified into hand-over-hand hauling mechanism. Our point is that the hand-over-hand hauling might occur randomly rather than sequentially among the subunits of the ClpB ring. To make our point clear, we have revised the manuscript as follows: “This might imply that ratchet-like substrate translocation is exerted randomly rather than sequentially among the subunits of the ClpB ring.” (page 20, line 6 from bottom). We also think that transitions between open and closed rings may be required for threading of a single polypeptide chain into the pore of the ClpB ring. This point has been clarified in the revised manuscript (page 20, line 4 from bottom). Furthermore, according to the comment raised by the Reviewer #1, we have deleted the following sentence from the revised manuscript: “Alternatively, interaction with the substrate peptide chain is required to exert the sequential rotary mechanism.”

Reviewer #3 (Remarks to the Author):

The paper has been significantly improved and is ready for publication.

Response: We appreciate recommendation of publication by the Reviewer #3.